# An immobilized antibody-based affinity grid strategy for on-grid purification of target proteins enables high-resolution cryo-EM

Qiaoyu Zhao[1,5], Xiaoyu Hong[1,5], Yanxing Wang [1], Shaoning Zhang[2], Zhanyu Ding[1], Xueming Meng [1], Qianqian Song[1], Qin Hong [1], Wanying Jiang[3], Xiangyi Shi[4], Tianxun Cai[2] & Yao Cong [1,3] ✉

In cryo-electron microscopy (cryo-EM), sample preparation poses a critical bottleneck, particularly for rare or fragile macromolecular assemblies and those suffering from denaturation and particle orientation distribution issues related to air-water interface. In this study, we develop and characterize an immobilized antibody-based affinity grid (IAAG) strategy based on the high-affinity PA tag/NZ-1 antibody epitope tag system. We employ Pyr-NHS as a linker to immobilize NZ-1 Fab on the graphene oxide or carbon-covered grid surface. Our results demonstrate that the IAAG grid effectively enriches PA-tagged target proteins and overcomes preferred orientation issues. Furthermore, we demonstrate the utility of our IAAG strategy for on-grid purification of low-abundance target complexes from cell lysates, enabling atomic resolution cryo-EM. This approach greatly streamlines the purification process, reduces the need for large quantities of biological samples, and addresses common challenges encountered in cryo-EM sample preparation. Collectively, our IAAG strategy provides an efficient and robust means for combined sample purification and vitrification, feasible for high-resolution cryo-EM. This approach holds potential for broader applicability in both cryo-EM and cryo-electron tomography (cryo-ET).

With rapid advancements in instrumentation and data analysis methods/ software, cryo-electron microscopy (cryo-EM) single-particle analysis has emerged as a powerful tool for the structural characterization of biomolecules, particularly macromolecular complexes[1–6]. However, for low abundance and fragile macromolecular complexes, along with biomolecules susceptible to preferred orientation, disintegration, or aggregation issues linked to interference from the air-water interface (AWI), the purification and vitrification procedures have emerged as critical bottlenecks for cryo-EM[7–12]. A promising approach involves the utilization of affinity grids for on-grid specimen purification[13–17]. While initial applications have been limited in scope and resolution, with most resolutions limited to 8–40 Å[13,15,16,18–20] (with a recent exception of MS2 virions at a resolution of 3.03 Å[17]), it is conceivable that a highly specific and well-conjugated tag, coupled with a stringent on-grid

purification protocol, could serve as a powerful method to isolate and examine rare biological assemblies[21–23].

The affinity grid strategy involves modifying the surface of transmission electron microscopy (TEM) grids with an affinity layer. This layer extracts and concentrates target biological assemblies on the grid, keeping the target away from the air-water interface. With constant efforts over the past decade, several strategies have been developed for generating affinity layers on TEM grids. These strategies include: (1) Functionalized lipids monolayers. They are functionalized with Ni-NTA or high-affinity small molecules, allowing them to capture His-tagged or other target proteins[13,14,20,24,25]. (2) Streptavidin 2D crystals. These crystals can capture biotinylated proteins[26–28]. This strategy has recently been applied in G-quadruplex RNA-mediated PRC2 dimer formation and cryo-EM

[1]Key Laboratory of RNA Innovation, Science and Engineering, Shanghai Institute of Biochemistry and Cell Biology, Center for Excellence in Molecular Cell Science, Chinese Academy of Sciences, University of Chinese Academy of Sciences, 200031 Shanghai, China. [2]State Key Laboratory of High Performance Ceramics and Superfine Microstructure, Shanghai Institute of Ceramics, Chinese Academy of Sciences, 200050 Shanghai, China. [3]Key Laboratory of Systems Health Science of Zhejiang Province, School of Life Science, Hangzhou Institute for Advanced Study, University of Chinese Academy of Sciences, Hangzhou, China. [4]Shanghai Nanoport, Thermo Fisher Scientific, Shanghai, China. [5]These authors contributed equally: Qiaoyu Zhao, Xiaoyu Hong. ✉e-mail: cong@sibcb.ac.cn

structure determination[29]. However, these two strategies require special handling and/or tools as well as user experience for grid preparation, and may introduce substantial background noise. The latter strategy requires computational removal of the 2D lattice in Fourier space. (3) Antibody-based affinity grids. These grids can concentrate target biomolecules on grid[16,30,31]. However, the antibodies are placed on the grid surface without chemical immobilization, which may not withstand multiple rounds of washing to remove contaminations. (4) Functionalized carbon[32,33], graphene[34,35], and graphene oxide (GO) surfaces[15,17,36,37]. These can introduce bioactive ligands onto supporting-film covered cryo-EM grid, enabling the enrichment of target proteins[15,17,32,33,36,37] or overcome preferred orientation problems[34,35]. However, this strategy requires freshly homemade carbon/graphene/GO supporting films conjugated with sufficient oxygen contents to chemically decorate the surface[15,17,32,34–37], or customized chemical reagents to introduce biological ligands[15,33,36,37]. Despite these advancements, there is a need for a more efficient and robust cryo-EM affinity grid strategy to perform on-grid purification, especially for low-abundance or fragile target complexes, and to address preferred orientation problems.

In a previous study, a novel PA–NZ-1 epitope tag system was developed[38]. This system consists of the NZ-1 antibody possessing high affinity toward the dodecapeptide dubbed PA tag, and is characterized by slow dissociation kinetics. This system has been used in subunit identification within macromolecular complexes[39–43]. Building upon the PA–NZ-1 affinity system, here we developed an immobilized antibody-based affinity grid (termed IAAG) strategy for high-resolution cryo-EM. Utilizing 1-Pyrenebutyric acid N-hydroxysuccinimide ester (Pyr-NHS) as a linker, we immobilize the antibody or Fab fragment onto the grid surface. Our IAAG strategy has been proven effective in concentrating low-abundance complexes and overcoming preferred orientation problems. We further demonstrated the utility of the IAAG strategy in the on-grid purification of apoferritin and the CCT6 homo-oligomeric ring (CCT6-HR) complex from cell lysates, facilitating atomic resolution cryo-EM reconstructions. Moreover, it is convenient to apply, eliminating the need for customized GO/chemical reagents. Overall, the IAAG strategy shows great potential for on-grid purification and cryo-EM investigation of challenging macromolecular complexes at atomic resolution.

## Results
### IAAG strategy development
In our approach, we employ Pyr-NHS as a linker to immobilize NZ-1 Fab to GO/carbon grid surfaces. Pyr-NHS is a chemical crosslinker that can noncovalently anchor onto carbon/graphene/GO-covered grid surfaces (sharing $sp^2$-hybridized carbon lattice) via π-π interaction through its pyrene moiety[44]. This retains the $sp^2$ lattice of the carbon skeleton without potentially disrupting the surface structure and electrical conductivity[45,46]. Concurrently, Pyr-NHS has a bioactive ester head that can react with primary amines in the NZ-1 Fab or other antibody/biomolecules, forming stable amide bonds to immobilize them on the grid surface (Fig. 1a). Pyr-NHS has been used in efficient graphene-based biosensor systems[47–52]. For efficient affinity grid preparation, we opted for a commercially available GO-coated cryo-EM grid (abbreviated as GO grid) to assemble our IAAG grid. Furthermore, we used only the Fab fragment of the NZ-1 antibody to reduce background noise.

### IAAG grid assembly and surface property evaluations
To assemble the IAAG grid, we first modified the GO surface by immersing the GO-coated grid in Pyr-NHS solution for 1 hour. We then washed the grid with Dimethylformamide (DMF) (Fig. 1b). To evaluate the efficiency of Pyr-NHS modification on the GO film, we employed X-ray photoelectron spectroscopy (XPS). The elemental analysis of Nitrogen N 1s from the XPS data revealed a new component emerged on the Pyr-NHS treated GO surface, which was absent in the untreated one (Fig. 2a). This observation confirms the successful loading of Pyr-NHS onto the GO film.

Subsequently, we applied NZ-1 Fab (75 μg/ml) onto the Pyr-NHS-treated GO grid and incubated it at room temperature for 45 minutes. The

grid was then washed with Tris buffer to terminate the crosslink reaction (Fig. 1b). Atomic force microscopy (AFM) scanning of the grid surface revealed that the IAAG grid surface became apparently more uneven compared with the control GO surface (Fig. 2b), indicating successful immobilization of NZ-1 Fab on the IAAG grid surface. Additionally, we examined the Pyr-NHS treated GO grid and the control untreated GO grid before and after applying NZ-1 Fab, using negative staining electron microscopy (NS-EM). This analysis demonstrated that both grid surfaces appeared plain prior to NZ-1 Fab addition (Fig. 2c). In contrast, post-application, the Pyr-NHS modified GO grid exhibited abundant immobilized NZ-1 Fab (Fig. 2d), while the control grid showed a very sparse distribution of Fab. Collectively, these data confirm the high immobilization efficiency of Pyr-NHS to NZ-1 Fab on the GO surface through the cross-linking reaction.

### IAAG can enrich target protein and overcome preferred orientation problems for atomic resolution cryo-EM
Low concentration and preferred orientation issues often impede high-resolution cryo-EM for macromolecular complexes[9,22,53]. To tackle these challenges, we first examined whether the IAAG strategy could enrich target proteins on the grid. As a test sample, we employed a low concentration of purified PA-tagged apoferritin, with the PA tag inserted at the exposed N-terminus of apoferritin. This was evident from the sparse presence of apoferritin on the control GO grid (Fig. 3a). In contrast, the IAAG grid displayed an obviously higher quantity of apoferritin (Fig. 3b), enabling us to determine an apoferritin cryo-EM structure at 2.5 Å resolution (Fig. 3c, Table 1, Supplementary Fig. 1a–c). These data underscore the effectiveness of the IAAG strategy in enriching target proteins for atomic resolution cryo-EM structural studies.

Pyr-NHS immobilizes NZ-1 Fab (or other antibodies/Fabs) by crosslinking with the primary amine in lysine residue or their N-terminus. The abundance of lysine residues on the outer surface of NZ-1 Fab (Fig. 3d) allows Pyr-NHS to immobilize NZ-1 Fab in diverse orientations. Therefore, we investigated the efficacy of our IAAG strategy in mitigating preferred orientation problems. In a recent study[54], we encountered preferred orientation issues while studying the homo-oligomeric ring (HR) complex formed by the CCT6 subunit of the yeast group II chaperonin TRiC/CCT. To assess our strategy, we employed CCT6-HR as a sample and inserted a PA tag into a surface-exposed loop in the middle portion of CCT6 (Fig. 3e). The PA peptide, known to form a β-turn in the antigen-binding pocket of the NZ-1 antibody[39], allowed us to insert it into various exposed turn-forming loops of target proteins[40].

We then used CCT6-HR in the presence of nucleotide analog ATP-AlFx, termed CCT6-HR-ATP-AlFx, as a testing sample. Our control experiment with CCT6-HR-ATP-AlFx confirmed the dominant distribution of top view particles (~87.2%), but a severe deficiency of side views (12.8%) (Fig. 3f). The subsequent 3D reconstruction revealed a map exhibiting severe stretch features, and the Euler angle distribution confirmed scarcity of side views, further validating the preferred orientation issue (Fig. 3g). In contrast, by utilizing IAAG grid, the raw image and reference-free 2D class averages exhibited diverse views, notably including previously missing side and tilted side views (which increased to 49.8%) (Fig. 3h). The Euler angle distribution showed the particles distributed more rationally (Fig. 3i), enabling us to determine the cryo-EM structure of CCT6-HR-ATP-AlFx at a resolution of 2.6 Å (Table 1, Supplementary Fig. 1d–g). These tests collectively demonstrate that our IAAG strategy can effectively overcome preferred orientation problems for atomic resolution cryo-EM.

### IAAG strategy for on-grid purification of TBCA-apoferritin from cell lysates for high-resolution cryo-EM
Low-abundance and fragile macromolecular complexes often suffer from tedious and time-consuming purification processes for structure studies. The ultimate goal of affinity grid strategy is to establish an efficient and robust on-grid purification method for target proteins directly from cell lysates for high-resolution cryo-EM, especially addressing rare and fragile assemblies.

In this study, we investigated the feasibility of our IAAG strategy for on-grid purification of PA tag-inserted target complexes from cell lysates for high-resolution cryo-EM. We constructed a PA-tagged TBCA-apoferritin display system, drawing inspiration from strategies that involve presenting a small protein on a cage-like structure, such as apoferritin[55] or DARPin cage[56–58]. In our system, we fused a TBCA protein (13 kDa) to the N-terminus of an apoferritin subunit (21 kDa). The expectation was that each of the 24 apoferritin subunits would display a TBCA molecule (Fig. 4a). Our SDS-PAGE and NS-EM analyses suggested that the cell lysates of PA-tagged TBCA-apoferritin contained the target complex along with numerous impurity proteins (Supplementary Fig. 2a, b, Supplementary Fig. 6). To perform on-grid purification (Fig. 1c), we immersed the freshly assembled IAAG grids in the cell lysates containing TBCA-apoferritin for 20 minutes, allowing the targets to bind to the grids.

Subsequently, after washing the grids three times with protein buffer to remove nonspecifically bound proteins, we immediately cryo-froze the IAAG grid with on-grid purified samples, which are ready for cryo-EM data acquisition. It appears that the IAAG strategy can effectively purify and enrich tagged TBCA-apoferritin from cell lysates (Fig. 4b). In contrast, the control GO grid failed to do so, displaying only sparsely distributed apoferritin and numerous contaminant proteins from cell lysates (Fig. 4c).

To analyze the distribution of TBCA-apoferritin within ice, we performed cryo-electron tomography (cryo-ET) analysis on the IAAG sample. It appeared that the immobilized NZ-1 Fab indeed anchored target proteins near the GO grid surface (Fig. 4d, Supplementary Movie. 1). The interaction between Fab and target proteins keeps the proteins away from the AWI, thereby avoiding protein denaturation and preferred orientation problems

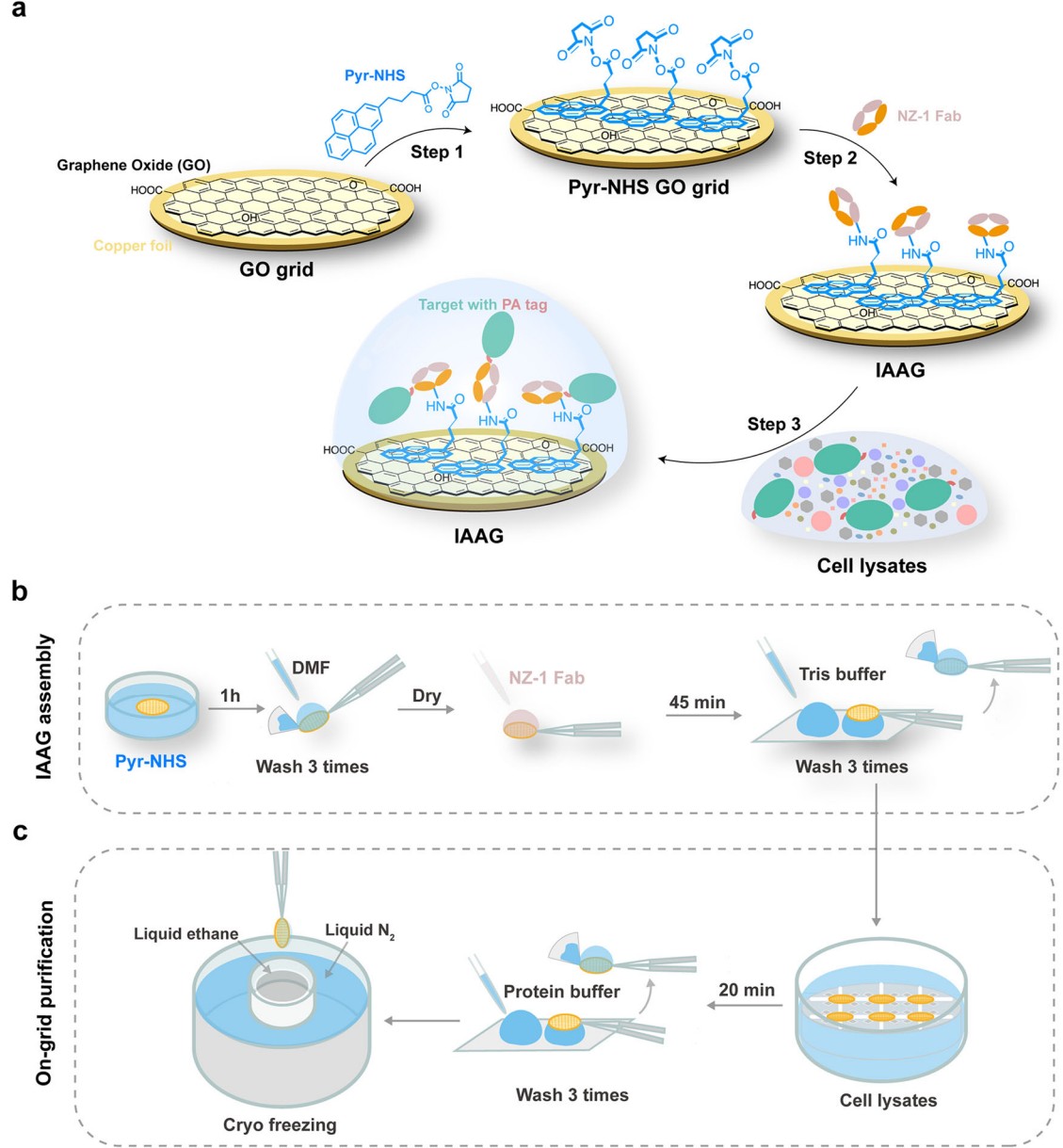

**Fig. 1 | Schematic illustration of the IAAG strategy. a** Schematic illustration of the assembly of the IAAG grid and the on-grid purification of target proteins for cryo-EM. **b** Workflow for the assembly of the IAAG grid. First, the GO-covered cryo-EM grid is immersed in Pyr-NHS soluted in DMF for 1 hr, followed by washing with DMF. Subsequently, NZ-1 Fab is added onto the grid surface for 45 minutes and then washed by Tris buffer to terminate the crosslink reaction. **c** Procedure for on-grid purification of target proteins form cell lysates by IAAG grid, followed by immediate cryo-freezing. The freshly assembled IAAG grids were immersed in cell lysates containing PA-tagged target for 20 minutes, followed by washing three times with protein buffer to remove nonspecifically bound proteins. The grids were immediately cryo-frozen.

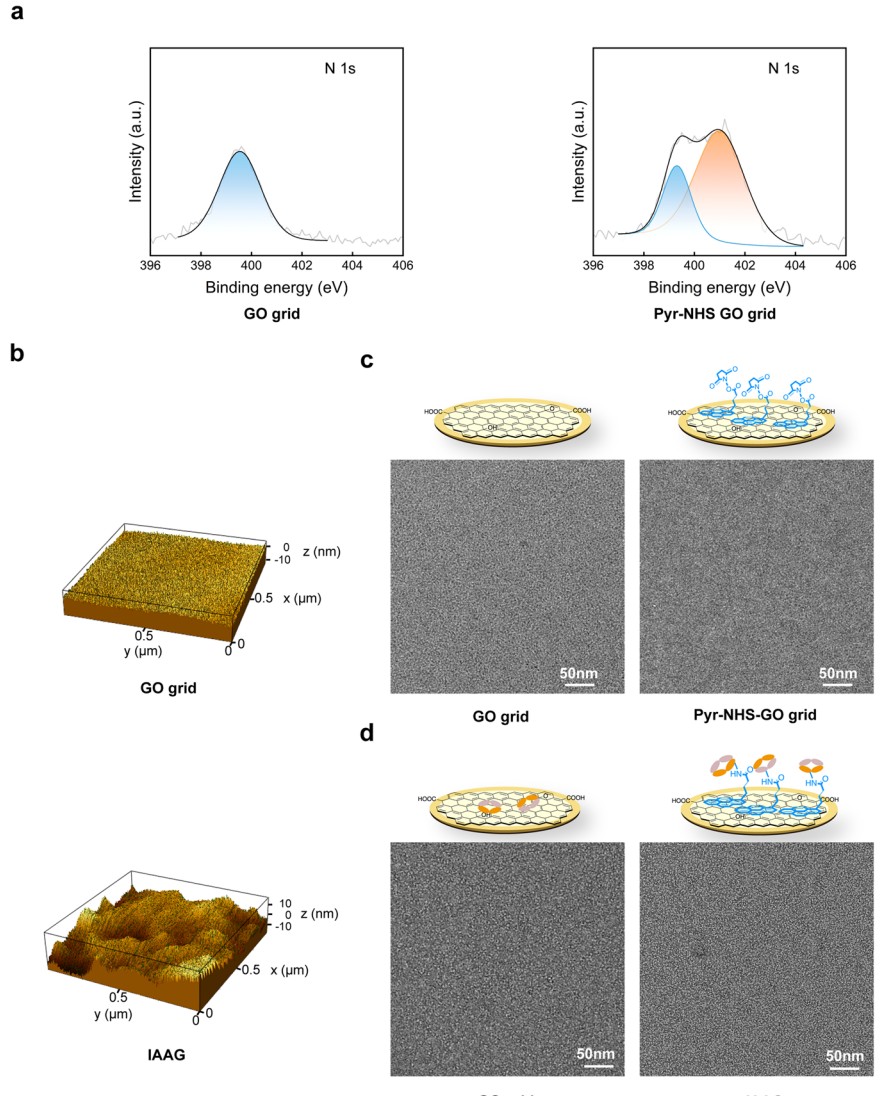

**Fig. 2 | Surface property evaluation of the IAAG grid. a** XPS spectra of the N 1 s of the GO grid (left) and the Pyr-NHS treated GO grid (right). The light gray line represents the original data while the black line presents the fit. The blue peak represents the N 1 s state of the untreated commercial GO grid, and the orange color scheme represents the new bond state of N 1 s on the Pyr-NHS treated GO surface. **b** AFM mapping of the GO grid (top) and the IAAG grid (bottom). **c** NS-EM image of GO grid (left) and Pyr-NHS treated GO grid (right). **d** NS-EM image of the GO grid (left) and the IAAG grid (right) treated with NZ-1 Fab. It appears that many more Fab densities (small white dots) can be observed on the IAAG grid.

in the vitrification process. These observations underscore the promise of the IAAG strategy in addressing common challenges in cryo-EM sample preparation.

We subsequently determined a consensus cryo-EM map of TBCA-apoferritin that was on-grid purified using the IAAG strategy, achieving a nominal resolution of 2.4 Å (Table 1, Supplementary Fig. 2c–e). The apoferritin portion of the TBCA-apoferritin map was better resolved, revealing atomic resolution structural features (Supplementary Fig. 2f). However, the TBCAs situated outside the apoferritin appear intrinsic dynamic and were less well resolved (Supplementary Fig. 2d, g). To improve the structural features of TBCA, we performed symmetry expansion with each duplicated particles rotated into one of the O symmetry-related positions[59,60]. After focused classification on this position in conjunction with 3DVA, we obtained a TBCA-focused refined map at the nominal resolution of 2.8 Å (Fig. 4e, Supplementary Fig. 2c, e). It displays improved density for the C-terminal helix of TBCA, which immediately fused to the N-terminal loop of apoferritin, at a local resolution of 4.0–5.0 Å (Fig. 4e, f). Nevertheless, the peripheral region of TBCA remained elusive, likely due to its dynamic nature. While further

engineering to tether TBCA more rigidly to the apoferritin scaffold could constrain its movements and facilitate high-resolution structure determination, such modifications are beyond the scope of the present study. Putting together, our data demonstrate the feasibility of the IAAG strategy for on-grid purification of target biomolecules from cell lysates and for high-resolution cryo-EM.

## IAAG strategy for on-grid purification of CCT6-HR from cell lysates

We further evaluated the on-grid purification ability of the IAAG strategy when dealing with target complexes with low expression levels. The yeast CCT6-HR complex was used as a test system (Supplementary Fig. 3a), which exhibited a rather low expression level in *E.coli*. The fractions containing the CCT6-HR were barely identifiable in the Coomassie blue-stained SDS-PAGE result (Supplementary Fig. 3b, Supplementary Fig. 7), and were hardly detected in the NS-EM (Supplementary Fig. 3c). The ring-shaped CCT6-HR complex was only sparsely observed in fractions 20-22 (Supplementary Fig. 3c), which were collected for on-grid purification using the IAAG strategy.

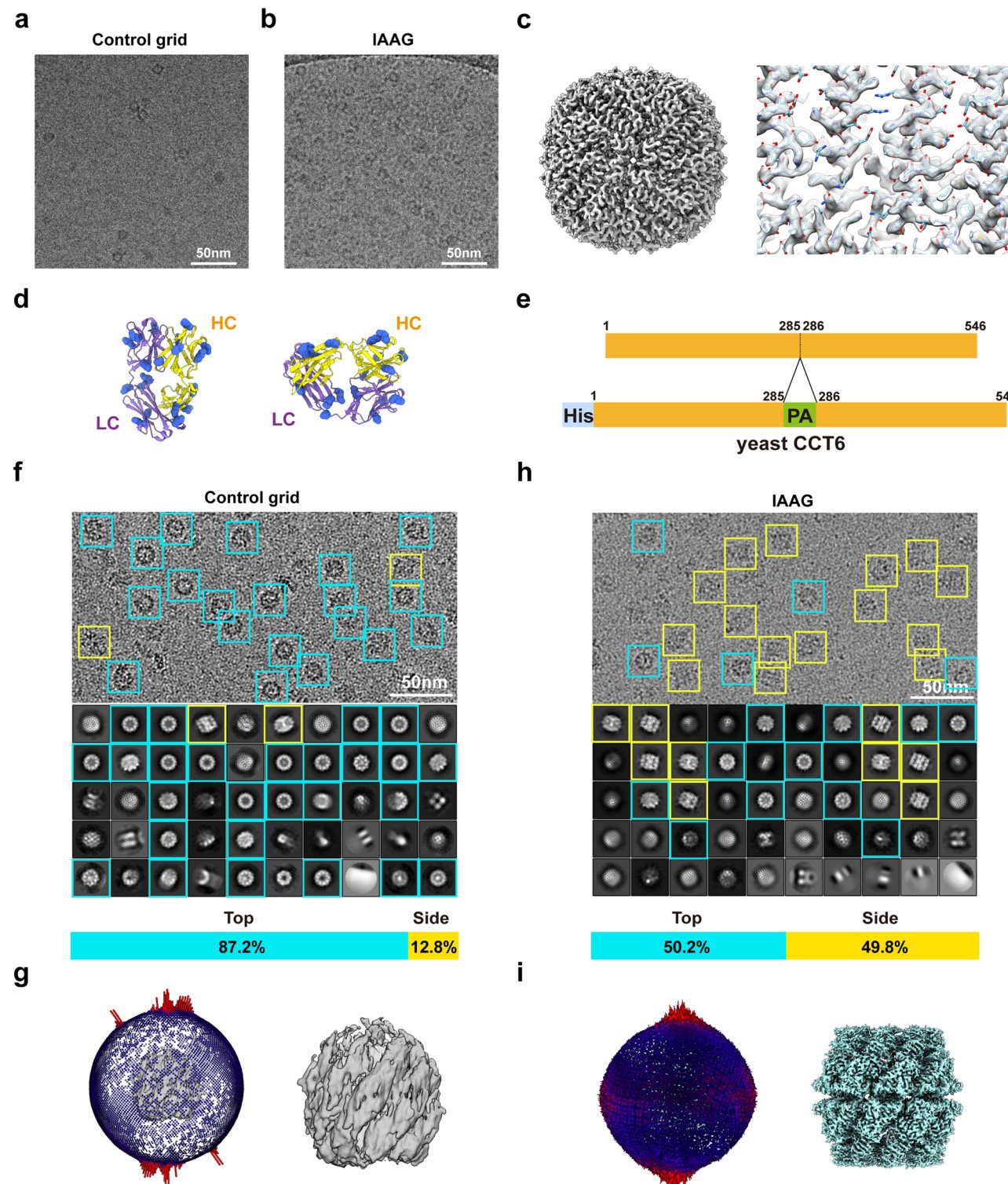

**Fig. 3 | IAAG can enrich target protein and overcome preferred orientation problems for atomic resolution cryo-EM. a** Cryo-EM image of the PA-tagged apoferritin prepared by the control GO grid, showing the low concentration of the target. **b** Cryo-EM image of the same sample prepared with IAAG grid, suggesting enrichment of the target protein. (**c**) 2.5-Å-resolution cryo-EM map of the IAAG enriched apoferritin (left), and high-resolution structure features (right) (model PDB: 6Z6U). **d** Side and top views of the NZ-1 Fab structure (PDB: 4YNY), with the lysine residues colored in cornflower blue. **e** Diagram of the His tag and PA tag inserted yeast CCT6 plasmid construct. **f** Cryo-EM image, reference-free 2D class averages, top/side view particle population analysis for the CCT6-HR-ATP-AlFx on

control Quantifoil grid. This demonstrated obvious preferred top-view orientation problems. Top/tilted top views are indicated by cyan square, and side/tilted side views in the yellow square. This color scheme was followed in the rest of this figure. **g** The corresponding 3D reconstruction (no imposed symmetry) and particle Euler angle distribution for the CCT6-HR-ATP-AlFx on the control Quantifoil grid. **h** Cryo-EM image, reference-free 2D class averages, top/side view particle population analysis for the CCT6-HR-ATP-AlFx on the IAAG grid. **i** The corresponding 3D reconstruction (no imposed symmetry) and particle Euler angle distribution for the CCT6-HR-ATP-AlFx on the IAAG grid, which helps overcome the preferred orientation problems.

## Table 1 | Cryo-EM reconstruction statistics

| | Apoferritin (EMDB-38143) | CCT6-HR-ATP-AlFx (EMDB-38142) | TBCA-apoferritin-consensus (EMDB-38145) | TBCA-focused refined TBCA-apoferritin (EMDB-39651) | CCT6-HR (EMDB-38147) |
|---|---|---|---|---|---|
| **Data collection** | | | | | |
| Microscope | Titan Krios | | | | Glacios |
| Camera | K2 | K3 | | | Falcon 4i |
| Magnification | 29,000 | 81,000 | | | 130,000 |
| Voltage (kV) | 300 | 300 | | | 200 |
| Electron exposure (e⁻/Å²) | 41 | 56.8 | | | 50 |
| Defocus range (µm) | −1.0 ~ −2.0 | −0.6 ~ −2.5 | | | −0.6 ~ −2.5 |
| Pixel size (Å) | 0.81 | 0.86 | | | 0.89 |
| **Reconstruction** | | | | | |
| Software | RELION 3.1 | RELION 3.1 & cryoSPARC 4.2.1 | | | |
| Symmetry imposed | O | D8 | O | C1 | D8 |
| Initial particle images (no.) | 107,179 | 440,563 | 145,772 | 898,560 | 391,997 |
| Final particle images (no.) | 43,720 | 144,370 | 37,440 | 109,152 | 83,979 |
| Map resolution (Å) | 2.5 | 2.6 | 2.4 | 2.8 | 2.8 |
| Map resolution range (Å) | 2.4–3.0 | 2.5–3.0 | 2.2–4.0 | 2.5–5.0 | 2.5–3.5 |
| FSC threshold | 0.143 | | | | |

Due to the low abundance of CCT6-HR in cell lysates, we extended the incubation time of IAAG grid in cell lysates to 1 hour. In the cryo-EM raw image, a considerable number of CCT6-HR in distinct top/side views can be recognized from the IAAG sample, while barely any CCT6-HR could be identified in the control GO grid (Fig. 5a, b). This demonstrates the efficacy of the IAAG strategy in on-grid purification of low-abundance proteins in cell lysates. We determined the cryo-EM structure of CCT6-HR to a resolution of 2.8 Å (Fig. 5c, d, Table 1, Supplementary Fig. 3d, e). The structure exhibited high-resolution structure features (Fig. 5e). This underscores the effectiveness of our IAAG strategy in on-grid purification of low-abundance target complexes.

### Feasible grid surfaces for the IAAG grid assembly

Given that Pyr-NHS noncovalently anchors onto surfaces sharing sp²-hybridized carbon lattice, we hypothesize that the IAAG strategy could also be applied to carbon/graphene surfaces, in addition to the GO surface. We tested this by using a continuous carbon-covered grid to assemble the IAAG. Our test demonstrated that the IAAG grid using carbon supporting film successfully purified PA-tagged TBCA-apoferritin from cell lysates (Supplementary Fig. 4a), while the control grid failed to do so (Supplementary Fig. 4b). Since our IAAG strategy performed well on continuous carbon surfaces, which lack hydrophilic group disturbances, we anticipate that it will perform excel on graphene grids, potentially outperforming its performance on GO grids. This is because GO grids have hydrophilic groups distributed across their surface. Still, the feasibility of graphene remains to be examined. In conclusion, surfaces of GO/carbon/graphene, all sharing the sp²-hybridized carbon lattice, are suitable for assembling IAAG grids.

### Discussion

For low-abundance or fragile macromolecular assemblies, as well as systems suffering from the AWI-related preferred orientation issues in cryo-EM, the utilization of affinity grids for on-grid purification of target complexes directly from cell lysates represents a potent solution. In this study, we developed an efficient immobilized antibody-based affinity grids (IAAG) strategy (Fig. 1). This strategy employs Pyr-NHS as a linker to noncovalently anchor antibody/Fab or bait proteins onto the sp²-hybridized carbon

lattice of the grid surface, including GO, carbon, or graphene, through cross-linking with their lysine residues or N-terminus. Our findings demonstrate that the IAAG strategy effectively enriches target proteins (Fig. 3a, b), and overcome preferred orientation problems (Fig. 3f–i). Notably, it facilitates on-grid purification of low-abundance target complexes (Fig. 5), streamlining the purification process, reducing the demand for large amounts of biological samples, and enabling atomic resolution cryo-EM (Figs. 4, 5). Collectively, our IAAG strategy provides an efficient and robust approach for combined sample purification and vitrification, making it feasible for high-resolution cryo-EM.

In contrast to existing affinity grid strategies that predominantly rely on freshly homemade graphene or GO supporting film functionalized with customized chemical reagents[15,17,34–37], our IAAG strategy use commercially available GO/carbon/graphene grids, along with the readily accessible cross-linker Pyr-NHS. The assembly of IAAG grids typically takes less than 2 hours (Fig. 1b). Consequently, our strategy is characterized by enhanced user-friendliness, efficiency, and broad applicability, thereby making it accessible to non-specialist laboratories. Additionally, our further test revealed that the Pyr-NHS treated GO grid exhibited considerably enhanced enrichment of the purified CCT6-HR sample compared to both the holey carbon grid and the unmodified GO grid (Supplementary Fig. 5). These results underscore the potential of Pyr-NHS coated grids in immobilization and enrichment of purified proteins in cryo-EM structural studies, aligning with previous studies utilizing other Pyr derivatives for similar purposes[7,61].

Unlike previous methods that merely place Fab/antibody on the grid surface[16,30,31], the IAAG strategy immobilizes it through cross-linking with Pyr-NHS (Fig. 1). This enhances stability and effectiveness, enabling IAAG to withstand multiple rounds of grid washing, immersion in cell lysates, and vitrification. Moreover, in conjunction with the high affinity and slow dissociation kinetics of PA tag/NZ-1 antibody[38], IAAG facilitates on-grid purification of low-abundance target proteins from cell lysates. Additionally, the presence of multiple lysine residues randomly distributed across the Fab/antibody surface (Fig. 3d), allows it to anchor on the grid surface in diverse orientations. This coupled with the effective tethering of the target proteins away from the AWI by the Fab/antibody (Fig. 4d, Supplementary Movie. 1), collectively prevents the AWI-related disintegration, aggregation,

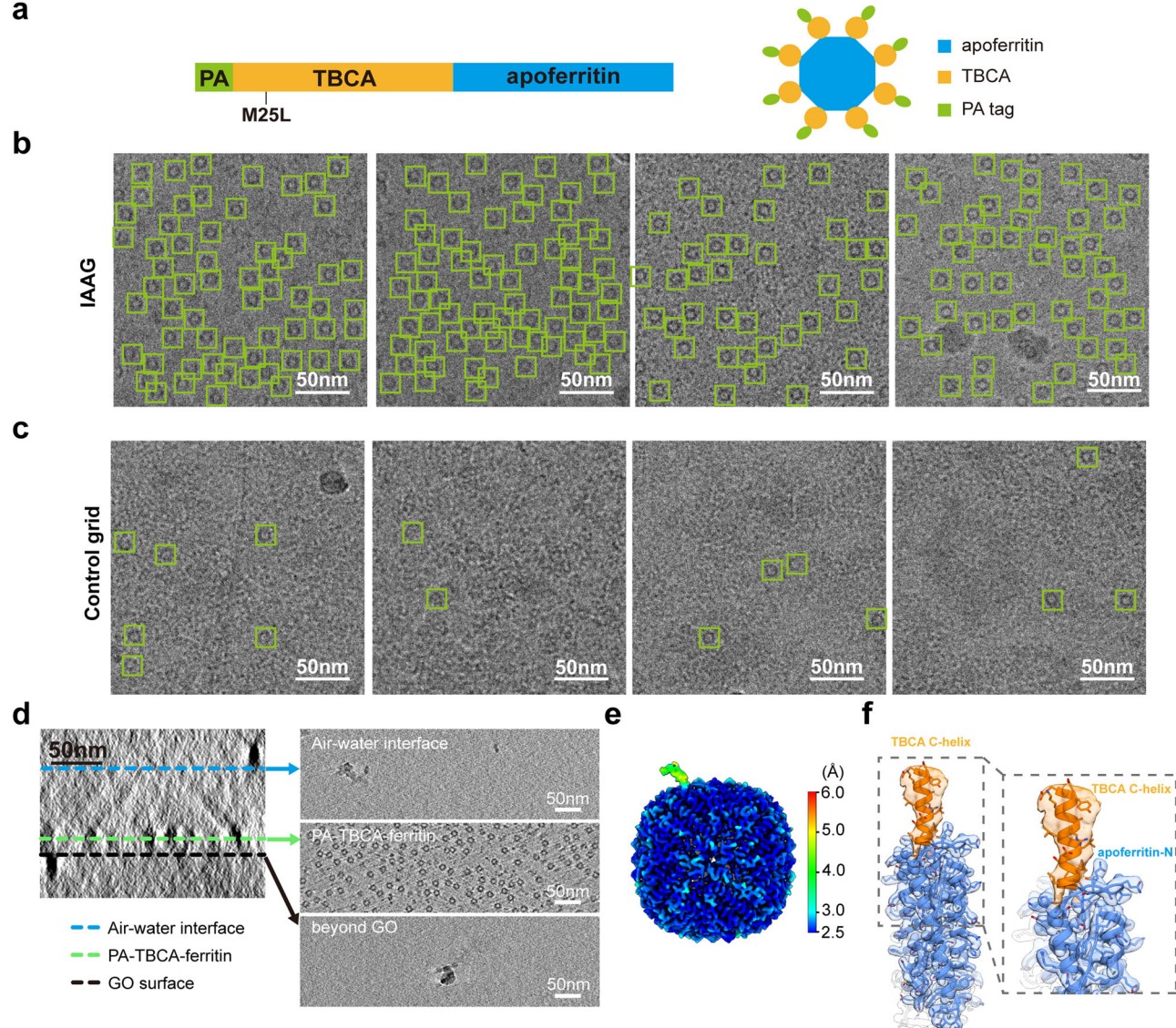

**Fig. 4 | Application of IAAG strategy in on-grid purification of TBCA-apoferritin from cell lysates for atomic-resolution cryo-EM. a** A diagram of the PA-TBCA-apoferritin plasmid construct. **b** Cryo-EM images of the IAAG on-grid purified PA-TBCA-apoferritin from cell lysates. Many targets (green squire) can be observed. **c** Cryo-EM images of the control GO grid treated with the cell lysates. Only very sparsely distributed target can be observed. **d** Cryo-ET analysis of the IAAG cryo sample, revealing that the TBCA-apoferritin particles are mostly distributed in a layer near the NZ-1 Fab treated GO surface and away from the air-water interface. The AWI is distinguished by the ice contaminations. **e** The local resolution map of the one TBCA position focus-refined TBCA-apoferritin map, at the normal resolution of 2.8 Å. **f** The model-map fitting of the TBCA C-terminal helix and an apoferritin subunit from the focus-refined map. The apoferritin is in cornflower blue, and the TBCA C-terminal helix in orange (apoferritin PDB: 6Z6U, TBCA model from AlphaFold2).

or preferred orientation issues. Notably, it appears that the Fab attached to the grid surface dose not impede atomic resolution cryo-EM structure determinations for apoferritin and CCT6-HR (with molecular weights ranging from ~450–1000 kDa). Still, it remains to be tested whether a full-length IgG might introduce background interference that could hinder atomic resolution reconstruction.

Moreover, the versatility of the IAAG strategy extends to the utilization of other high-affinity pairs, such as an antibody/Fab/nanobody with a specific antigen. More broadly, it has the potential to selectively enrich or on-grid purify target proteins directly from endogenous resources, guided by a specific antibody or other high-affinity binding bait protein. This eliminates the need for special engineering on target proteins. Consequently, the IAAG strategy presents itself as a viable alternative for cryo sample preparation of fragile assemblies. Furthermore, IAAG on-grid purification holds the potential to preserve weak interaction partners and offer a more complete conformational landscape under

endogenous conditions. Such valuable information may be compromised with conventional preparation processes, which typically rely on traditional chromatography purification in conjunction with filtration-based concentration steps[11]. Collectively, our IAAG strategy exhibits broader applicability and holds promise for diverse applications in cryo-EM research.

In summary, we developed the immobilized antibody-based affinity grid (IAAG) strategy. This approach utilizes Pyr-NHS as a linker to immobilize antibody/Fab/nanobody on carbon/graphene/GO-coated cryo-EM grids, indicating its versatility and wide applicability. The IAAG strategy facilitates on-grid purification of tagged complexes directly from cell lysates, enabling atomic resolution cryo-EM. Moreover, the Pyr-NHS linker holds the potential to immobilize various other antibody/nanobody or bait proteins, thus enabling on-grid affinity purification of endogenous target biomolecules or organelles without the need for an affinity tag. This development paves the way for a more user-friendly and robust approach

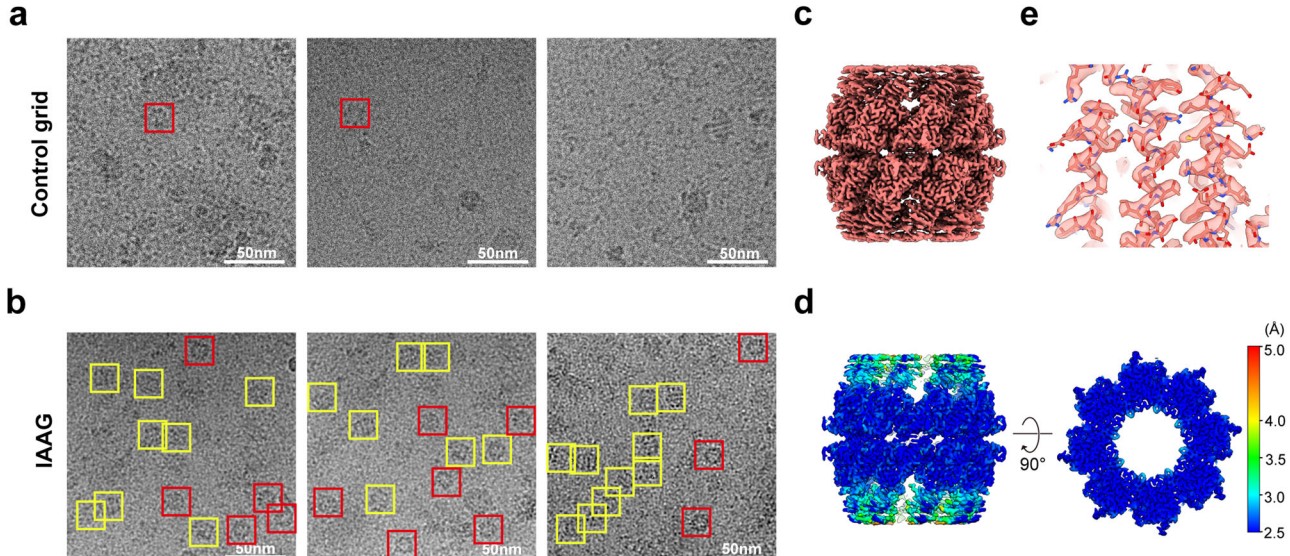

**Fig. 5 | IAAG on-grid purified CCT6-HR from cell lysates for high-resolution cryo-EM. a** Cryo-EM images of the control GO grid treated with CCT6-HR cell lysates, indicating barely any CCT6-HR could be identified in the control grid. **b** Cryo-EM images of the IAAG on-grid purified CCT6-HR from cell lysates. Obviously, more CCT6-HR particles in distinct top/side views can be recognized.

Here top view indicated by a red square, and the side view is in a yellow square. **c** The 2.8-Å-resolution cryo-EM map of CCT6-HR reconstructed from the IAAG data. **d** Local resolution estimation of the CCT6-HR complex. **e** High-resolution structure features of the CCT6-HR complex (PDB: 5GW5 subunit CCT6).

with broader applicability in cryo-EM structural studies, and potential extensions to cryo-ET in situ structure studies.

## Materials and Methods
### Plasmid construction
To construct the PA-tag-containing apoferritin for the purified sample, the sequence of human ferritin H chain was cloned into the pGEX2T vector with a tandem GST tag, thrombin site, TEV site, and a PA tag inserted at the N terminus of ferritin. For the on-grid purification, the sequence of ferritin was cloned into the pETDuet-1 vector with only a PA tag fused at the N terminus of ferritin.

The full-length human TBCA gene (NM_004607) was amplified by PCR using total cDNAs from HEK293 cells. The PA-TBCA-apoferritin construction was performed by fusing a PA tag to the N-terminus of TBCA and apoferritin to the C-terminus of TBCA, which was then cloned into the pET28a vector. To reduce the relative flexibility between TBCA and apoferritin, the first 4 amino acids of the N terminus of apoferritin were truncated. To circumvent the problem that expression of TBCA cDNA results in a truncated form due to internal initiation at Met25, we performed site-directed mutagenesis of Met25 to Leu25 (Fig. 4a).

For yeast CCT6-HR, the PA tag was inserted between the site of C285 and G286 following our previous study[40]. A 6xHis tag was added at the N terminus of CCT6 for affinity purification, and then constructed into the pET28a vector (Fig. 3e). For the preparation of CCT6-HR cell lysates for on-grid purification, the N-terminal 6xHis tag was removed (Supplementary Fig. 3a) from the plasmid construction.

### Protein expression and purification
All of the above plasmids were transformed into *E.coli* BL21 (DE3) respectively for subsequent IPTG-induced protein expression. The *E.coli* cultures were grown in LB medium until they reached an $OD_{600}$ of 1.2. IPTG was then added to induce protein expression at 16 °C for 18 h. All cell cultivation conditions were kept consistent.

The centrifuged pellet for apoferritin was resuspended in a lysis buffer (1×PBS, 0.5% Triton X-100, 1 mM DTT, protease inhibitors (Roche)). The lysate was centrifuged at 20,000 $g$ for 1 h, and the supernatant was incubated with Glutathione Sepharose for 1.5 h at 4 °C. The resin was recovered using a gravity chromatography column and washed with a wash buffer

(1×PBS, 0.5% Triton X-100, 1 mM DTT). Apoferritin was eluted using an elution buffer (50 mM Tris-HCl pH 7.5, 100 mM NaCl, 0.5 mM EDTA, 1 mM DTT, 10 mM glutathione). The pooled eluate containing apoferritin was concentrated using Millipore Amicon Ultra centrifugal filter units (30-kDa cutoff).

For the purification of the yeast CCT6-HR, the cells were lysed in a buffer containing 20 mM HEPES-KOH pH 7.4, 300 mM NaCl, 10 mM $MgCl_2$, 5 mM β-mercaptoethanol, 10% glycerol, 2 mM ATP. One EDTA-free complete protease inhibitors EDTA free (Roche) was added in 100 mL lysis buffer. The lysate was centrifuged at 18,000 rpm for 30 minutes followed by 200,000 $g$ for 1.5 h. The supernatant was then incubated with Ni-Beads (EMD Millipore Corporation) at 4 °C for 30 minutes. The CCT6 single subunit was eluted off with 300 mM imidazole, and the eluent was concentrated using an Amicon Ultra 100 kDa MWCO centrifugal filter (Millipore). The concentrated protein was further separated by 10–40% (w/v) glycerol gradient centrifugation. The fractions containing the CCT6-HR were combined and concentrated using an Amicon Ultra 100 kDa MWCO centrifugal filter (Millipore).

### Pretreatment of cell lysates
For the cell lysates used in IAAG on-grid purification, the centrifuged cell pellet was resuspended in lysis buffer. The lysate was subjected to centrifugation at 18,000 rpm for 30 minutes, followed by ultra-centrifugation at 150,000 g for 2 h. The supernatant was then concentrated using a Millipore Amicon Ultra centrifugal filter unit (100-kDa cutoff). The supernatant of the lysate was then subject to glycerol density gradient centrifugation with a 10-40% (w/v) glycerol gradient to approximately separate proteins with different sedimentation coefficients.

### Purification and preparation of NZ-1 Fab
The NZ-1 Fab was prepared as described in a previous report[39], with some modifications. Briefly, the purified NZ-1 IgG was incubated with papain (300:1 W/W) in a PBS buffer (in the presence of 20 mM L-cysteine and 1 mM EDTA) for 3 h at 37 °C. The reaction was then quenched with 20 mM iodoacetamide. Subsequently, the Fab was purified by running it over a HiTrap SP FF column (GE Healthcare) that had been pre-equilibrated with 10 mM PBS.

## Assembly of the IAAG grid

To assemble the IAAG grid, Graphene Oxide-lacey carbon grids (300 mesh, EMR) or Ultrathin carbon-lacey grids (400 mesh, TED PELLA, INC) were immersed in 1 mM Pyr-NHS (Santa Cruz Biotechnology, Inc.) soluted by DMF for 1 h at room temperature. The grids were then washed three times with DMF to remove the excess cross-linker, and blotted with filter paper to remove remained liquid. After drying for 10 minutes at room temperature, the grids were ready for antibody coating. NZ-1 Fab was diluted to 75-80 μg/mL in phosphate-buffered saline (PBS) pH 7.4. Then, 5 μL of the NZ-1 Fab was deposited onto the grid and incubated for 45 minutes at room temperature. The grids were then washed three times with Tris buffer (50 mM Tris-HCl pH 7.5, 100 mM NaCl, 0.5 mM EDTA, 1 mM DTT) with 30 μl per time, and blotted with filter paper. At this point, the IAAG grid has been assembled and is ready for immediate use.

## Cryo-EM sample preparation using IAAG grid and the IAAG on-grid purification

For the enrichment of the apoferritin sample (Fig. 3a–c), the IAAG grids were incubated with 0.27 mg/ml PA-tagged apoferritin at room temperature for 20 minutes. The grids were then washed four times with Tris buffer. Right before typical cryo freezing, a volume of 2 μl buffer was placed onto the grid, which was blotted with a Vitrobot Mark IV (Thermo Fisher Scientific) and then plunged into liquid ethane cooled by liquid nitrogen.

To prepare the sample of yeast CCT6-HR in the presence of 1 mM ATP-AlFx, the purified CCT6-HR was incubated in a buffer containing 1 mM ATP, 5 mM MgCl$_2$, 5 mM Al (NO$_3$)$_3$, and 30 mM NaF at 30 °C for 1 h. Then 2.2 μl of CCT6-HR-ATP-AlFx was deposited directly onto the IAAG grid and vitrified as described above using Vitrobot Mark IV.

For on-grid purification of target proteins from cell lysates, the IAAG grids were incubated with the cell lysates for 20 minutes for TBCA-apoferritin, and 60 minutes for CCT6-HR at room temperature. The grids were then washed three times with Tris buffer. Right before typical cryo freezing, a volume of 2.2 μl buffer was placed onto the grid, followed by plunge freezing utilizing Vitrobot Mark IV.

## Cryo-EM data acquisition

Images for IAAG enriched apoferritin, the yeast CCT6-HR-ATP-AlFx, and the on-grid purified TBCA-apoferritin samples were acquired on a Titan Krios transmission electron microscope (Thermo Fisher Scientific) operated at 300 kV. The images of purified apoferritin were recorded on a K2 Summit direct electron detector (Gatan) in super-resolution mode, with a pixel size of 0.405 Å/pixel. Each movie was dose-fractioned into 36 frames, with a total accumulated dose of 41 e$^-$/Å$^2$ on the specimen. All images were collected by utilizing SerialEM[62], with final defocus values ranging from −1.0 to −2.0 μm. Cryo-EM movies of the on-grid purified TBCA-apoferritin and the yeast CCT6-HR-ATP-AlFx were recorded on a K3 direct electron detector (Gatan) operated in the counting mode, yielding a pixel size of 0.86 Å. This was done under a low-dose condition in an automatic manner using EPU software (Thermo Fisher Scientific). Each frame was exposed for 0.1 s, and the total accumulation time was 3 s, leading to a total accumulated dose of 56.8 e$^-$/Å$^2$ on the specimen (Table 1).

Cryo-EM movies of the on-grid purified CCT6-HR sample were acquired on a Glacios electron microscope (Thermo Fisher Scientific) operated at 200 kV. The movies were recorded on a Falcon 4i direct electron detector (Thermo Fisher Scientific) operated in the counting mode (yielding a pixel size of 0.89 Å) using EPU software. The total accumulated dose is 50 e$^-$/Å$^2$ in total accumulation time of 4.42 s on the specimen.

## Image processing and 3D reconstruction

Single-particle analysis was performed using RELION 3.1[63] and cryoSPARC 4.2.1[64]. All images were aligned and summed using MotionCorr2 whole-image motion correction software[65]. After CTF parameter determination using CTFFIND4[66], particle auto-picking was performed by utilizing crYOLO 1.7.6[67].

For the determination of the IAAG-enriched apoferritin structure (Supplementary Fig. 1a), a total of 762 micrographs were used. After particle auto-picking, manual particle checking, and two rounds of reference-free 2D classification, 81,675 particles remained. After two subsequent rounds of 3D classification, 43,720 particles with good features were used for further refinement, resulting in a map at 2.5 Å resolution.

For the CCT6-HR-ATP-AlFx complex (Supplementary Fig. 1d), a total of 440,563 particles were picked using crYOLO. Following 2D classification, 224,246 good particles remained. These particles underwent one round of 3D classification, followed by a no-align 3D classification with C1 symmetry, resulting in the selection of 144,370 particles with good features. Subsequently, these particles were subjected to two rounds of CTF refinement and Bayesian Polishing. Then refinement was conducted with C1/D8 symmetry, yielding reconstructions at resolutions of 3.2 Å/2.6 Å, respectively.

For the data processing of TBCA-apoferritin (Supplementary Fig. 2c), 145,772 particles were picked from 1799 micrographs. After 2D classification, 47,270 good particles remained. These particles underwent 3D classification with O symmetry, resulting in the selection of 37,440 particles with good features. These particles were used to reconstruct a map at 3.2 Å resolution. After two rounds of CTF refinement and Bayesian Polishing, a consensus map at 2.4 Å resolution was reconstructed. Due to the flexibility of TBCA relative to the apoferritin scaffold, the TBCA density around the apoferritin could only be seen at a lower threshold (Supplementary Fig. 2d). To improve the structural features of TBCA, we performed symmetry expansion in CryoSPARC[64], by duplicating the original particles' poses by 24-fold, with each duplicated particles rotated into one of the O symmetry-related positions. We then conducted focused classification of this TBCA position in conjunction with 3D variability analysis (3DVA). After this focused classification, 316,064 particles exhibiting clear TBCA density contributed to the reconstruction of a 2.7 Å resolution map. Notably, the density corresponding to the TBCA C-terminal helix, immediately connected to the N-terminal loop of apoferritin, was better resolved. To further improve its quality, we applied a tighter mask on this region and conducted a second round of focused classification via 3DVA. Ultimately, we achieved a TBCA-focused refined TBCA-apoferritin map at 2.8 Å resolution, constructed from 109,152 particles, with the local resolution of the TBCA C-terminal helix ranging from 4~5 Å.

For the data processing of on-grid purified CCT6-HR (Supplementary Fig. 3d), a total of 391,997 particles were picked by using crYOLO. After 2D classification, 129,678 good particles remained. These particles underwent 3D classification with C1 symmetry, resulting in the selection of 83,979 particles, which led to a C1 reconstruction at 4.0 Å resolution. After subsequent CTF refinement and Bayesian Polishing, the particles were imported to cryoSPARC to perform non-uniform refinement with D8 symmetry, which led to a reconstruction at 2.8 Å resolution.

The resolution was accessed based on the gold-standard criterion with FSC at 0.143. All figures were rendered using UCSF Chimera[68] or UCSF ChimeraX[69].

## Cryo-ET analysis

Cryo-ET movies of the TBCA-apoferritin sample were collected on a Titan Krios electron microscope (Thermo Fisher Scientific) operated at an accelerating voltage of 300 kV. The microscope was equipped with an energy filter (Gatan) operated at a slit width of 20 eV. The movies were collected at a magnification of 26,000× using SerialEM[62], and recorded on a K3 direct electron detector (Gatan) in the counting mode, yielding a pixel size of 2.70 Å. Tilt-series were acquired from −51° to +51°, with 3° increments and two tilts per reversal using a dose-symmetric scheme[70]. The exposure time was 1 s or 1.5 s with 10 frames for each tilt angle, and the total dose was approximately 68 e$^-$/Å$^2$. These frames were further motion-corrected using Warp[71]. The AreTomo software was used to align and reconstruct the tomogram[72]. For analysis and visualization, the IMOD software was utilized[73].

## Grid surface property characterization

The composition characterization of N 1 s signal was conducted by X-ray photoelectron spectroscopy (ESCALAB 250Xi, Thermo Fisher Scientific) with Mg $K_\alpha$ X-ray (1253.6 eV) radiation and an energy resolution of 0.6 eV. To eliminate the effect of solvent, the Pyr-NHS was soluted in methanol rather than DMF for the sample used in XPS analysis. The distribution of NZ-1 Fab on the grid surface along the Z-axis was characterized using atomic force microscopy (Omegascope SL, AIST-NT) in a tapping mode at 1.0 Hz scanning speed with commercial tips (HQ:NSC14/Cr-Au, Mikro-Masch) in resonance frequency of 142.4 kHz.

## Statistics and reproducibility

No statistical method was used to predetermine the sample size for cryo-EM and cryo-ET data. The cryo-EM images of poor quality were deleted based on the defocus, astigmatism, and resolution for better reconstruction. Data collection and processing statistics were summarized in Table 1.

## Reporting summary

Further information on research design is available in the Nature Portfolio Reporting Summary linked to this article.

## Data availability

All data presented in this study are available within figures and in Supplementary Information. Cryo-EM maps have been deposited at the Electron Microscopy Data Bank with accession codes EMDB-38143 for apoferritin, EMDB-38142 for CCT6-HR-ATP-AlFx, EMDB-38145 for TBCA-apoferritin consensus map, EMDB-39651 for TBCA-focus refined TBCA-apoferritin map, and EMDB-38147 for CCT6-HR.

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

## Acknowledgements

We are grateful to the staff of the NCPSS Electron Microscopy facility, Database and Computing facility, and Protein Expression and Purification facility for instrument support and technical assistance. We also thank the Shanghai Nanoport, Thermo Fisher Scientific for their instrument support on Glacios and Tundra. This work was supported by grants from the Strategic Priority Research Program of CAS (XDB37040103 and XDB0570000), the NSFC (32130056 and 31872714), the National Key R&D Program of China (2017YFA0503503), and the Shanghai Pilot Program for Basic Research from CAS (JCYJ-SHFY-2022-008).

## Author contributions

Y.C., Q.Z. and X.H. developed the IAAG strategy with the involvement of Z.D. Q.Z., X.H., Y.W., Q.H., Q.S. and W.J. constructed the plasmids and purified the proteins. Q.Z. and X.H. prepared the cryo-samples, collected the cryo-EM data (X.S. was involved in data acquisition on Glacios), and performed the reconstruction. X.M. performed the cryo-ET. Q.Z. and S.Z. performed the

XPS. S.Z. and T.C. performed the AFM. Q.Z., X.H. and Y.C. analyzed the data. Q.Z. and Y.C. wrote the manuscript with the input of X.H. and Y.W.

## Competing interests

Center for Excellence in Molecular Cell Science is currently applying for a patent (application no. 2023111349814) covering the procedures of IAAG assembly and on-grid purification detailed in the manuscript and the applications on cryo-EM. The patent lists Y.C., Q.Z., X.H. and Y.W. as inventors. All authors, including these authors and others, declare no competing interests.
