## [Peer Review File · Communications Biology]

Reviewers' comments:

Reviewer #1 (Remarks to the Author):

This manuscript describes some strategies by which a particular linker, 1-Pyrenebutyric acid N-hydroxysuccinimide ester (Pyr-NHS), when adsorbed to a continuous, thin support film, can be used to fabricate affinity grids. As is appropriately summarized in this manuscript, affinity grids have the potential to improve the success rate of preparing "difficult" macromolecules for cryo-EM. The use of Pyr-NHS in the way described here is a novel and very promising approach, which will be of great interest to the cryo-EM community. The quality of EM data collection and analysis is first-rate. I would cite this work for sure.

MAJOR COMMENTS

1. Although I am very impressed with most of the other parts of this work, I feel that the attempt to tether a sub-50 kDa protein to the much larger apoferritin particle still needs improvement before it is appropriate to publish it. I thus recommend that the authors delete that part.
2. I also feel that the manuscript is often rather poorly/loosely organized, occasionally repetitive, and tedious to follow. I thus feel that major revision in how it is organized, avoiding unnecessary repetition, would improve the manuscript considerably. No additional experimental work is needed.

MINOR COMMENTS

I would like to see the result of one further "control" experiment: After binding Pyr-NHS to GO, please determine whether that reagent can be removed by washing with an organic solvent. This should be the case, if Pyr-NHS is bound to the GO support film in the way that is assumed by the authors (see the top of Figure 1). This is not such an important "control" to be included, however; it would just support the statement that the reagent is believed to be physisorbed via π - π interactions.

Little is included in this manuscript to document that graphene, as opposed to graphene oxide, can also be used. One would think that graphene would actually be superior to GO (which has many hydrophilic groups distributed over its surface), if the chemisorption mechanism shown at the top of Figure 1 is correct. I thus suggest that the authors say a bit more about why they do not show results obtained with graphene.

I would think that other investigators would try to immobilize purified protein complexes by reacting them directly with the Pyr-NHS coated grids. It would thus be helpful for the authors to mention whether that approach works, assuming that they had tried it themselves.

Reviewer #2 (Remarks to the Author):

Reviewer Comments: Communications Biology.

Zhao et al., describes an immobilised antibody based affinity grid (IAAG) for on-grid purification of target proteins and vitrification for cryo-EM data collection. The antibody was immobilized on GO/carbon grid surface. The paper is based on linking an antibody with a linker to the carbon surface of the grid.

Antibody based methods have been used earlier too. However, the work presented here is an

advancement of earlier method where the antibody is immobilized on the carbon/GO surface of an EM grid. The novelty proposed by the authors in this paper are :

1. The authors used Pyr-NHS as a linker to immobilize NZ-1 Fab to GO/carbon grid surfaces. This provides immobilised antibody based affinity grid (IAAG). They claim that this enhances stability and effectiveness, allowing IAAG to withstand multiple rounds of grid washing, immersion in cell lysates, and vitrification.
2. They show that IAAG strategy can help in alleviating preferred orientation bias for single particle cryo-EM. They suggest this was achieved, which may be due to the random orientation distribution of the immobilized NZ-1 Fab via the Pyr-NHS linker.
3. In addition this IAAG methods keeps the target particles away from Air Water Interface (AWI), which are usually the culprit for the preferred orientation of particles in single particle cryoEM.
4. The authors also demonstrated that their IAAG strategy can be utilized for on-grid purification of small tagged protein (TBCA-Apoferritin) and low-abundance target complexes (CCT6-HR complex) from cell lysates and enables atomic resolution cryo-EM structure determination.
5. The authors effectively demonstrate that all carbon/graphene/GO surfaces, all of which share the sp²-hybridized carbon lattice, are suitable for assembling IAAG grids.
6. The authors also claim that the IAAG grid assembly strategy presented in this paper typically takes less than 2 hours and thus is more user friendly, efficient, and broadly applicable, making it accessible to non-specialist laboratories.

The work complements other related work in literature, for example :

- a) A 3D-printed flow-cell for on-grid purification of electron microscopy samples directly from lysate (2023). <https://www.sciencedirect.com/science/article/pii/S104784772300062X>
 - b) Microfluidic protein isolation and sample preparation for high-resolution cryo-EM (2019) : <https://www.ncbi.nlm.nih.gov/pmc/articles/PMC6660723/>
 - c) Antibody-based affinity cryoEM grid (2016). doi: 10.1016/j.ymeth.2016.01.010
 - d) Single-step antibody-based affinity cryo-electron microscopy for imaging and structural analysis of macromolecular assemblies (2014) : <https://www.sciencedirect.com/science/article/abs/pii/S1047847714000926>
 - e) Strategy for the Use of Affinity Grids to Prepare Non-His-Tagged Macromolecular Complexes for Single-Particle Electron Microscopy (2010) <https://www.sciencedirect.com/science/article/abs/pii/S0022283610005371>
- etc.

The manuscript was a pleasure to read. It is well written and control experiments were done appropriately where ever needed. The manuscript can be accepted in its present form.

REVIEWER COMMENTS

Reviewer #1 (Remarks to the Author):

This manuscript describes some strategies by which a particular linker, 1-Pyrenebutyricacid N-hydroxysuccinimide ester (Pyr-NHS), when adsorbed to a continuous, thin support film, can be used to fabricate affinity grids. As is appropriately summarized in this manuscript, affinity grids have the potential to improve the success rate of preparing “difficult” macromolecules for cryo-EM. The use of Pyr-NHS in the way described here is a novel and very promising approach, which will be of great interest to the cryo-EM community. The quality of EM data collection and analysis is first-rate. I would cite this work for sure.

--We sincerely appreciate all the encouraging comments and insightful suggestions from this reviewer, which let us explore more thoroughly to improve our manuscript.

Major points:

Q1-1. Although I am very impressed with most of the other parts of this work, I feel that the attempt to tether a sub-50 kDa protein to the much larger apoferritin particle still needs improvement before it is appropriate to publish it. I thus recommend that the authors delete that part.

A1-1: We appreciate your valuable suggestion and apologized for any confusion. For this testing data, our primary objective is to assess the feasibility of the IAAG strategy for direct on-grid purification of target proteins from cell lysates. Regarding the testing sample, we utilized an apoferritin display system, wherein a TBCA protein (13 kDa) was fused to the N-terminus of each apoferritin subunit (21 kDa). The expectation was that each of the 24 apoferritin subunits would exhibit a displayed TBCA molecule, as illustrated in Fig. 4A (also provide as Fig. R1A for the convenience of the reviewer and editor). In the consensus map with imposed O symmetry (at 2.4 Å resolution), we indeed observed 24 extra densities corresponding to the displayed TBCAs outside the apoferritin “shell” when using a lower rendering threshold (Fig. R1B). However, these TBCA densities disappeared when the rendering threshold was higher (Fig. R1B), likely due to the lack of constraints between TBCA and the apoferritin scaffold and the resulted in dynamics of TBCA.

To improve the structural features of TBCA, we employed the symmetry expansion strategy in CryoSPARC¹, by duplicating the original particles' poses by 24-fold, with each duplicated particles rotated into one of the O symmetry-related positions. After two rounds of focused local classification on this position in conjunction with 3D variability analysis (3DVA), we obtained a TBCA focus-refined map at the nominal resolution of 2.8 Å (Fig. R1C). In this map, the C-terminal helix of TBCA, which immediately fused to the N-ternimal loop of apoferritin, was better captured, reaching a local resolution

of 4.0-5.0 Å (Fig. R1C, D). Nevertheless, the peripheral region of TBCA remained elusive, likely due to its dynamic nature. While further engineering to tether TBCA more rigidly to the apoferritin scaffold could constrain its movements and facilitate high-resolution structure determination, such modifications are beyond the scope of the present study. Given the enhanced structural features of the TBCA C-terminal helix portion, we intend to remain this data, while tuning down the tone on the apoferritin display system accordingly. We have updated the relevant content in both the Results and Methods sections in our revised manuscript (L. 209-220 on P. 8-9, L. 459-471 on P. 17-18), as well as Fig. 4E-F and Supplementary Fig. 2C-E.

Fig. R1 Single particle analysis of the IAAG on-grid purified TBCA-apoferritin from cell lysates.

(A) Diagram illustrating the PA-TBCA-apoferritin plasmid construct. (B) Consensus map of TBCA-apoferritin. Surface color is rendered based on the radius from the center, with the 24 TBCA densities visible at lower rendering threshold and disappearing at higher rendering threshold. (C) The local resolution estimation of the TBCA focus-refined map of TBCA-apoferritin, now at the nominal resolution of 2.8 Å. It displays improved density for the TBCA C-terminal helix at a local resolution of 4.0-5.0 Å. (D) The model-map fitting of the TBCA C-terminal helix and an apoferritin subunit from the focus-refined TBCA-apoferritin map.

Q1-2. I also feel that the manuscript is often rather poorly/loosely organized, occasionally repetitive, and tedious to follow. I thus feel that major revision in how it is organized, avoiding unnecessary repetition, would improve the manuscript considerably. No additional experimental work is needed.

A1-2: We have followed the suggestion from our reviewer to reorganize our manuscript and enhance its readability and flow. Briefly, we have moved some of the redundant panels from the original figures (original Fig. 3H-I, Fig. 4E-F) to supplementary figures (Supplementary Fig. 1F-G, Supplementary Fig. 2F-G) and removed the related contents. We have also relocated the IAAG grid assembly using continuous carbon to the last session of the Results, and reduced the discussion on the apoferritin display system, which is somewhat beyond the main topic of the current study. We hope that the current version is more concise and meets the standard of the reviewer.

Minor points:

Q1-3. I would like to see the result of one further “control” experiment: After binding Pyr-NHS to GO, please determine whether that reagent can be removed by washing with an organic solvent. This should be the case, if Pyr-NHS is bound to the GO support film in the way that is assumed by the authors (see the top of Figure 1). This is not such an important “control” to be included, however; it would just support the statement that the reagent is believed to be physisorbed via π - π interactions.

A1-3: We sincerely appreciate the valuable suggestion provided by the reviewer. We have followed the suggestion to perform a control experiment to address the removal of Pyr-NHS from the GO surface through an organic solvent washing. Specifically, we immersed the Pyr-NHS treated GO grid in methanol and subjected it to low-speed rotation, followed by additional rinsing by methanol (Fig. R2A). Subsequently, we evaluated the surface properties of both the Pyr-NHS GO grid and the Pyr-NHS removal GO grid using XPS analysis (Fig. R2B). As demonstrated by the elemental analysis of N 1s, the peak attributed to Pyr-NHS decreased significantly from 42.66% to 18.71% (Fig. R2B). This result confirms the effective removal of Pyr-NHS from the GO surface through organic solvent treatment, substantiating our hypothesis regarding the π - π interactions between Pyr-NHS and the GO surface. In addition, aromatic molecules have been widely employed for the non-covalent surface modifications of sp^2 -hybridized carbon derivatives, such as graphene and amorphous carbon, particularly in TEM studies²⁻⁵.

Fig. R2 Removal of Pyr-NHS from GO surface using organic solvent. (A) Workflow illustrating the Pyr-NHS treatment of the GO grid and subsequent Pyr-NHS removal. The Pyr-NHS treated GO grid was immersed in methanol and subjected to low-speed rotation three times, followed by three additional methanol rinses (1 ml of methanol per rinse) to wash out the physisorbed Pyr-NHS. (B) XPS spectra of the N 1s for the Pyr-NHS GO grid (left) and Pyr-NHS removal GO grid (right). The light gray line represents the original data, while the black line depicts the fitted data. The blue peak represents the N 1s state of the commercial GO grid, and the orange color scheme represents the bond state of N 1s resulting from Pyr-NHS treatment.

Q1-4. Little is included in this manuscript to document that graphene, as opposed to graphene oxide, can also be used. One would think that graphene would actually be superior to GO (which has many hydrophilic groups distributed over its surface), if the chemisorption mechanism shown at the top of Figure 1 is correct. I thus suggest that the authors say a bit more about why they do not show results obtained with graphene.

A1-4: Thanks for the good points. Our tests were conducted exclusively using GO and continuous carbon coated grids, which are readily available in the current market and widely utilized in the cryo-EM community than the graphene grid. Since our IAAG strategy performed well on continuous carbon surfaces without hydrophilic group disturbances, we anticipate that it will excel on graphene grids, potentially outperforming its performance on GO grids. This is because GO grids have many

hydrophilic groups distributed over its surface. We have included a brief discussion on this aspect in the revised manuscript (L. 249-253 on P. 10).

Q1-5. I would think that other investigators would try to immobilize purified protein complexes by reacting them directly with the Pyr-NHS coated grids. It would thus be helpful for the authors to mention whether that approach works, assuming that they had tried it themselves.

A1-5: Thanks for the insightful comments. The reviewer is right that the Pyr-NHS coated grids can indeed help to immobilize purified protein complexes. We employed the purified CCT6 homo-oligomeric ring (CCT6-HR) complex at low concentration as our testing sample. On the typical holey carbon grid (Cu R2/1, 200 mesh, Quantifoil), we observed minimal particle detection within the holes (Fig. R3A). When utilizing a GO covered grid, there was a slight increase in particle numbers (Fig. R3B). Strikingly, the Pyr-NHS treated GO grid exhibited significantly superior CCT6-HR enrichment (Fig. R3C), compared to both the Quantifoil holey carbon grid and the unmodified GO grid. These findings underscore the potential of Pyr-NHS coated grids in immobilization and enrichment of purified protein complexes in cryo-EM structural studies. We have now incorporated this content into our revised manuscript (L. 278-284 on P. 11, Supplementary Fig. 5).

Fig. R3 Application of Pyr-NHS coated grid for immobilization and enrichment of purified proteins. Purified CCT6-HR complex in the presence of ATP-AIFx (CCT6-HR-ATP-AIFx) was used as the testing sample. (A) Representative cryo-EM image of CCT6-HR-ATP-AIFx prepared using a Quantifoil holey carbon grid. (B) Cryo-EM image of CCT6-HR-ATP-AIFx on a GO grid. (C) Cryo-EM image of CCT6-HR-ATP-AIFx on a Pyr-NHS coated GO grid, illustrating a great enrichment of the particles using this type of grid. These observations demonstrate the effectiveness of Pyr-NHS coated GO grids in immobilizing and enriching purified protein complexes for cryo-EM studies.

Reviewer #2 (Remarks to the Author):

Reviewer Comments: Communications Biology.

Zhao et al., describes an immobilised antibody based affinity grid (IAAG) for on-grid purification of target proteins and vitrification for cryo-EM data collection. The antibody was immobilized on GO/carbon grid surface. The paper is based on linking an antibody with a linker to the carbon surface of the grid.

Antibody based methods have been used earlier too. However, the work presented here is an advancement of earlier method where the antibody is immobilized on the carbon/GO surface of an EM grid. The novelty proposed by the authors in this paper are:

1. The authors used Pyr-NHS as a linker to immobilize NZ-1 Fab to GO/carbon grid surfaces. This provides immobilised antibody based affinity grid (IAAG). They claim that this enhances stability and effectiveness, allowing IAAG to withstand multiple rounds of grid washing, immersion in cell lysates, and vitrification.
2. They show that IAAG strategy can help in alleviating preferred orientation bias for single particle cryo-EM. They suggest this was achieved, which may be due to the random orientation distribution of the immobilized NZ-1 Fab via the Pyr-NHS linker.
3. In addition this IAAG methods keeps the target particles away from Air Water Interface (AWI), which are usually the culprit for the preferred orientation of particles in single particle cryoEM.
4. The authors also demonstrated that their IAAG strategy can be utilized for on-grid purification of small tagged protein (TBCA-Apoferritin) and low-abundance target complexes (CCT6-HR complex) from cell lysates and enables atomic resolution cryo-EM structure determination.
5. The authors effectively demonstrate that all carbon/graphene/GO surfaces, all of which share the sp^2 -hybridized carbon lattice, are suitable for assembling IAAG grids.
6. The authors also claim that the IAAG grid assembly strategy presented in this paper typically takes less than 2 hours and thus is more user friendly, efficient, and broadly applicable, making it accessible to non-specialist laboratories.

The work complements other related work in literature, for example:

- a) A 3D-printed flow-cell for on-grid purification of electron microscopy samples directly from lysate (2023). <https://www.sciencedirect.com/science/article/pii/S104784772300062X>
- b) Microfluidic protein isolation and sample preparation for high-resolution cryo-EM (2019): <https://www.ncbi.nlm.nih.gov/pmc/articles/PMC6660723/>
- c) Antibody-based affinity cryoEM grid (2016). doi: 10.1016/j.ymeth.2016.01.010
- d) Single-step antibody-based affinity cryo-electron microscopy for imaging and structural analysis of macromolecular assemblies (2014): <https://www.sciencedirect.com/science/article/abs/pii/S1047847714000926>

e) Strategy for the Use of Affinity Grids to Prepare Non-His-Tagged Macromolecular Complexes for Single-Particle Electron Microscopy (2010)

<https://www.sciencedirect.com/science/article/abs/pii/S0022283610005371> etc.

The manuscript was a pleasure to read. It is well written and control experiments were done appropriately where ever needed. The manuscript can be accepted in its present form.

--We deeply appreciate the positive and encouraging comments on our manuscript provided by the reviewer, and the time and effort in reviewing our work. We are optimistic that our IAAG strategy will serve as a valuable tool for the cryo-EM community, particularly in handling challenging macromolecular systems.

Reference:

- 1 Punjani, A., Rubinstein, J. L., Fleet, D. J. & Brubaker, M. A. cryoSPARC: algorithms for rapid unsupervised cryo-EM structure determination. *Nature methods* **14**, 290-296, doi:10.1038/nmeth.4169 (2017).
- 2 Pantelic, R. S., Fu, W. Y., Schoenenberger, C. & Stahlberg, H. Rendering graphene supports hydrophilic with non-covalent aromatic functionalization for transmission electron microscopy. *Appl Phys Lett* **104**, doi:10.1063/1.4870531 (2014).
- 3 Scherr, J. *et al.* Noncovalent Functionalization of Carbon Substrates with Hydrogels Improves Structural Analysis of Vitrified Proteins by Electron Cryo-Microscopy. *ACS Nano* **13**, 7185-7190, doi:10.1021/acsnano.9b02651 (2019).
- 4 D'Imprima, E. *et al.* Protein denaturation at the air-water interface and how to prevent it. *Elife* **8**, doi:10.7554/eLife.42747 (2019).
- 5 Georgakilas, V. *et al.* Functionalization of graphene: covalent and non-covalent approaches, derivatives and applications. *Chem Rev* **112**, 6156-6214, doi:10.1021/cr3000412 (2012).

REVIEWERS' COMMENTS:

Reviewer #1 (Remarks to the Author):

The authors have revised their manuscript in a satisfactory way, in response to my initial comments.